# Nemaline Myopathy in Brazilian Patients: Molecular and Clinical Characterization

**DOI:** 10.3390/ijms231911995

**Published:** 2022-10-09

**Authors:** Juliana Gurgel-Giannetti, Lucas Santos Souza, Guilherme L. Yamamoto, Marina Belisario, Monize Lazar, Wilson Campos, Rita de Cassia M. Pavanello, Mayana Zatz, Umbertina Reed, Edmar Zanoteli, Acary Bulle Oliveira, Vilma-Lotta Lehtokari, Erasmo B. Casella, Marcela C. Machado-Costa, Carina Wallgren-Pettersson, Nigel G. Laing, Vincenzo Nigro, Mariz Vainzof

**Affiliations:** 1Department of Pediatrics, Service of Neuropediatrics, Federal University of Minas Gerais, Belo Horizonte 31270-901, MG, Brazil; 2Human Genome and Stem Cell Research Center, Department of Genetics and Evolutionary Biology, Biosciences Institute, University of São Paulo, São Paulo 05508-220, SP, Brazil; 3Rede Mater dei de Saúde, Belo Horizonte 30110-062, MG, Brazil; 4Departamento de Neurologia, Hospital das Clinicas da Universidade de São Paulo (USP), São Paulo 05403-000, SP, Brazil; 5Departamento de Neurologia, Hospital São Paulo, Universidade Federal de São Paulo (UNIFESP), São Paulo 04024-002, SP, Brazil; 6Folkhälsan Research Center, Department of Medical Genetics, Medicum, University of Helsinki, 00100 Helsinki, Finland; 7Children’s Institute, Hospital das Clínicas HCFMSUP, Faculdade de Medicina, Universidade de São Paulo, São Paulo 05508-220, SP, Brazil; 8Escola Bahiana de Medicina e Saúde Pública (EBMSP), Salvador 40110-909, BA, Brazil; 9Centre for Medical Research, Queen Elizabeth II Medical Centre, Harry Perkins Institute of Medical Research, University of Western Australia, Nedlands, WA 6009, Australia; 10Italy and Telethon Institute of Genetics and Medicine (TIGEM), Università della Campania Luigi Vanvitelli Napoli, 80078 Pozzuoli, Italy

**Keywords:** nemaline myopathy, nebulin, acta1, congenital myopathy

## Abstract

Nemaline myopathy (NM), a structural congenital myopathy, presents a significant clinical and genetic heterogeneity. Here, we compiled molecular and clinical data of 30 Brazilian patients from 25 unrelated families. Next-generation sequencing was able to genetically classify all patients: sixteen families (64%) with mutation in *NEB*, five (20%) in *ACTA1*, two (8%) in *KLHL40*, and one in *TPM2* (4%) and *TPM3* (4%). In the *NEB*-related families, 25 different variants, 11 of them novel, were identified; splice site (10/25) and frame shift (9/25) mutations were the most common. Mutation c.24579 G>C was recurrent in three unrelated patients from the same region, suggesting a common ancestor. Clinically, the “typical” form was the more frequent and caused by mutations in the different NM genes. Phenotypic heterogeneity was observed among patients with mutations in the same gene. Respiratory involvement was very common and often out of proportion with limb weakness. Muscle MRI patterns showed variability within the forms and genes, which was related to the severity of the weakness. Considering the high frequency of *NEB* mutations and the complexity of this gene, NGS tools should be combined with CNV identification, especially in patients with a likely non-identified second mutation.

## 1. Introduction

Nemaline myopathy (NM) is one of the most common structural congenital myopathies, with a significant clinical and genetic heterogeneity. NM estimated incidence is about 1:50,000 [1,2,3]. The histopathological marker is the presence of rod or nemaline bodies inside muscle fibers associated with type I fiber predominance. The spectrum of clinical presentation varies from severe onset in the neonatal period to juvenile/adult onset with mild weakness [2,4]. Patients usually show neonatal hypotonia and proximal, facial and axial weakness, which can be associated with skeletal deformities. Depending on the degree of muscle weakness, severity of the muscle weakness and age at onset, NM was clinically divided into six forms by the ENMC Nemaline Consortium [1]. More recently, a new classification was suggested with the aim to simplify and to give some information about the prognosis of NM patients. This classification includes the following forms: severe (intrauterine onset), typical (perinatal onset), mild (childhood or juvenile onset), distal, childhood onset with slowness, recessive *TNNT1* and other forms with unusual presentation [5].

Nowadays, more than 15 genes are related to NM, including *TPM3*, *NEB*, *ACTA1*, *TPM2*, *TNNT1*, *KBTBD13*, *CFL2* (COFILIN2), *KLHL40*, *KLHL41*, *LMOD3*, *MYO18B*, *MYPN*, *RYR3*, *TTN3*, *ADSSL1*, Filamin C and *MYH2*. Most of these genes encode structural or regulatory proteins associated with the thin filament in the skeletal muscle fiber [5].

NM can present autosomal dominant (AD) or autosomal recessive (AR) inheritance, with a high proportion of sporadic dominant cases [2,3]. The first genes identified associated with NM were the genes encoding slow α-tropomyosin (*TMP3*) [6,7], skeletal muscle α-actin 1 (*ACTA1*) [8] and nebulin (*NEB*) [9], all proteins of the muscle thin filament. Subsequently, mutations in other genes were described, including beta-tropomyosin (*TPM2* [10], slow troponin T (*TNNT1*) [11], cofilin-2 (*CFL2*) [12]. More recently, with the advance of large-scale molecular screening, mutations in several other genes were identified, including *KBTBD13* [13], *KLHL40* [14], *KLHL41* [15], *LMOD3* [16,17], *MYO18B* [18], myopalladin (*MYPN*) [19], *RYR3* [20], *TTN3* [21], *ADSSL1* [22], Filamin C [23] and *MYH2* [24] genes.

Among these known genes, mutations in the nebulin gene (*NEB*) are the most common cause of autosomal recessive NM, corresponding to around 50% of the cases [3]. This gene is huge, with 183 exons spanning 249 kb of the genomic sequence, and encodes the nebulin protein of approximately 600–900 kDa [25,26,27]. Therefore, the molecular study of this gene was a great challenge in previous years. More recently, with the advance of next-generation sequencing (NGS) methodologies, the screening of the *NEB* gene has been applied in larger cohorts of NM patients, and more mutations have been identified. Additionally, large structural variations have also been observed in this gene, indicating the necessity to associate molecular methods with copy number variation analyses [28,29].

Magnetic resonance imaging (MRI) has been considered a powerful method for the study of muscle diseases. Muscle MRI images allow the identification of patterns of muscle involvement according to the form of myopathy and can sometimes be suggestive of a specific gene mutation, mainly in patients with congenital myopathies, including nemaline myopathy [30,31,32,33].

Over the last 20 years, we have been screening for causative mutations in Brazilian patients with nemaline myopathy. We first used Sanger sequencing of genomic DNA for small genes, such as *ACTA1*, *TPM2* and *TPM3*, and the nebulin gene was screened in collaboration with NM Consortium members. Later, we introduced the use of next-generation sequencing, which allowed us to study all the known NM genes. Now, we were able to compile the molecular and clinical data of this cohort of 30 Brazilian patients and to correlate the molecular forms with clinical severity, findings in muscle biopsy and muscle MRI.

## 2. Results

### 2.1. Clinical Data

A total of 30 patients, 15 males and 15 females, from 25 unrelated families were evaluated. Five families presented more than one affected patient, one of them with a clear autosomal dominant inheritance (Family 7) and four with autosomal recessive form (two affected siblings: Families: 10, 16, 20, 23). The remaining 20 families presented with sporadic cases.

Patients were classified based on the severity of the disease: 24 with the typical form, 3 with the mild form and 3 with the severe neonatal form (Table 1) [5].

All patients who underwent muscle biopsy presented rods in muscle fibers. The distribution of rods inside muscle fibers was predominantly a mixed pattern of large subsarcolemmal rods associated with small intermyofibrillar rods (Figure 1).

Among the 30 patients, 20 individuals (P1, P2, P3, P5, P6, P7, P8, P10, P14, P15, P16, P17, P20, P21, P23, P24, P25, P26, P29, P30) were followed over months to years (six months to ten years) in a neuromuscular outpatient clinic by multidisciplinary teams.

Serial cardiac evaluations performed in these 20 patients over a period of up to ten years did not indicate any cardiac abnormalities.

Three patients died with respiratory complications (P10, P20 and P26).

### 2.2. Molecular Data

We identified pathogenic mutations in NM-related genes in all 25 studied families. *NEB* variants were present in 20 patients (P9–P28) from 16 families (all patients had two *NEB* variants). Five families showed heterozygous mutations in the *ACTA1* gene (P1–P5). In addition, in four families, mutations in the following genes were found: *TPM2* (P6), *TPM3* (P7/P8) and *KLHL40* (P29, P30) (Table 1).

#### 2.2.1. Patients with Mutations in the Nebulin Gene

Among the 20 patients (16 families) with *NEB* mutations, 17 individuals were classified as typical (P10–P22, P25, P26) and 3 with the mild form (childhood/juvenile form) (P9, P23, P24).

All families showed two nebulin mutations, including point mutations, small deletions or duplication and CNVs (P9–P28). In two families, there were two affected siblings (P11/P12 and P23/P24) carrying homozygous mutations, and family 10 (P11/P12 was consanguineous. In the other 14 families, the cases were compound heterozygous for two different *NEB* variations, including family 16 and 23 with two siblings. Parental segregation analysis performed in three cases confirmed the presence of the two mutations in trans (P14, P16, P17).

Initially, in two families (P21 and P22), we found a mutation in only one allele of the *NEB* gene, but using next-generation piplines for CNV identification, we found the second mutation: duplication of exons 82 to 105 in P21 and deletion of exon 29 in P22. Additional analysis using Motor Chip v3 [41] confirmed the duplication in patient 21, and parental segregation analysis showed that the CNV was de novo.

Among the total of twenty-five different identified variants in the *NEB* gene, ten were in the splice sites, nine were frameshift, three stop-gain, one non-frameshift duplication and two CNVs. Eleven of these variants were newly described mutations: four frameshift (c.8501delA, c.24304_24305+2T>A; c.23878_23881dup, c.3648delT), three splice site (c.20466+2T>A, c.19102-8_19102-4del; c.25405-1G>C), one stop-gain (c.16423A>T), one non-frameshift duplication (c.6869_6870insTGC) and two CNVs (exons 82–105 duplication, exon 29 deletion).

Four variants were recurrent: c.24579G>C in three families (P14, P16, P17), c.22170G>A; p.Tyr7390* in two families (P14 and P17), c.24189_24192dup; p.Glu8065Serfs*5 in two families (P10, P25); and c.5343+5G>A in two families (P23/P24 in homozygous and P27/P28 in heterozygous).

Although mutation c.24579G>C was detected in three patients from different families, they were from the same state of Brazil. Among these three patients, two (P14, P17) showed the same and previously described mutations in compound heterozygosity (c.22170C>A and c.24579G>C).

The follow-up of patients with *NEB* mutations could be performed in 11 individuals with the typical form (cases: P10, P14, P15, P16, P17, P20, P21, P23, P24, P25, P26). Among these patients, three died suddenly because of respiratory complications (P10, P20, P26). The age of death for P10, P20 and P26, was, respectively, 25 years, 15 years and 5 years of age.

Although three patients (P10, P21, P25) were classified as the typical form, they showed more intense weakness and needed ventilatory support and gastrotomy in the first two years of life. P10 presented with repetitive pneumonia and, at 18 months of age, he had tracheostomy and needed mechanical ventilation 24 h/day, and he died at 25 years of age due to respiratory complications. P21 had tracheostomy at the age of 8 months, during a respiratory infection. At the age of 5 years, the tracheostomy could be replaced by non-invasive ventilation (NIV). At her last clinical evaluation, she was 12 years old, still using gastrostomy and NIV for more than 16 h/day. P25 started using NIV at 7 months of age and is still using it for more than 16 h/day. He developed a progressive weakness, losing the capacity to control the head and to sit and needed a gastrostomy at 5 years old. These three patients did not achieve the capacity to walk, and all presented severe scoliosis.

Four patients (P14, P15, P16, P17) were using NIV, which was started during puberty. Two patients (P14 and P17) were using it during nighttime, and two patients were using it for more than 16 h each day (P15 and P16). All these patients were able to walk. However, one lost this ability at 17 years old (P16). In the first years of life, P16 showed proximal weakness in the upper and lower limbs, compatible with the typical form. During puberty, she developed an intense and progressive involvement of distal muscles in the lower limbs, which impaired her capacity to walk during adolescence. The weakness progression was accompanied by respiratory failure and severe scoliosis. At the age of 15, she was admitted for spinal surgery but had persistent severe respiratory failure, needing NIV almost 24 h a day.

P23 and P24, with the mild form of NM, are siblings and have shown a mild proximal and facial weakness with a stable course until now and are able to walk. They did not have ventilatory insufficiency or scoliosis at the last evaluation, at the age of 27 and 25 years, respectively.

#### 2.2.2. Patients with Mutations in the *ACTA1* Gene

Five patients had mutations in the *ACTA1* gene, four with the typical (P1, P2, P4, P5) and one with the severe form (P3).

All five presented heterozygous mutations in the *ACTA1* gene, and four of them were previously described missense variants: c.541G>C, p.Asp181His; c.478G>A, p.Gly160Ser; c.593G>A, p.Arg198His; and c.854T>G, p. Met285Arg.

In P2, we found a novel splice site intronic mutation (c.130-5T>A), classified as a variant of uncertain significance (VUS), following the ACMG pathogenicity classification guidelines, but it showed some pathogenic indicatives based on in silico prediction programs, including the Human Splicing Finder and MaxEntScan. This mutation was not present in the father or the mother and was considered a de novo mutation.

The follow-up of patients with *ACTA1* mutations was performed in four patients, three with the typical (P1, P2 and P5) and one with the severe form (P3).

Two patients (P2, P5) with the typical form did not need ventilatory support and were able to walk until the last evaluation. They did not present scoliosis; however, P5 was followed until the beginning of her puberty, and P2 had his last evaluation at 13 years old. One patient (P1) walked at 5 years of age but started using a wheelchair before 11 years of age and needed NIV in adult life.

P3 presented severe weakness since birth. He uses ventilatory support 24 h a day with tracheostomy and never acquired the capacity to walk. He presented an asymmetric weakness distribution. He showed a more intense weakness on the left side of the body and on the right side of the face. At the age of 13 years, he needed spinal surgery to correct a severe scoliosis.

#### 2.2.3. Patients with Mutations in Other Genes Associated with NM

One patient, P6, showed the recurrent mutation p.K7del in the *TPM2* gene and presented the typical form. He developed mild scoliosis during puberty and had been followed without the necessity for surgical procedure. He does not need ventilatory support. The clinical and genetic data were previously described [36].

Two patients from the same family (P7 and P8) showed a previously described mutation (c.391C>T; p.Arg131Cys) in the *TPM3* gene. Both patients presented with the typical form. P7 was 19 years old at the last evaluation; he developed a progressive scoliosis during puberty and needed NIV ventilation during nighttime. At the age of 15 years, he had scoliosis surgery. His father, P8, is 62 years old and started using NIV at the age of 51. Both have stable motor function and are still able to walk.

Two patients (P29, P30) showed two identical heterozygous mutations in the *KLHL40* gene. In patient 29, a segregation (c.1405G>A; p.Gly469Ser; c.1498C>T; p.Arg500Cys) study in the parents revealed that the previously described p.Gly469Ser mutation was present in the father [14]. The p.Arg500Cys was observed in the mother and is a novel mutation. Patient 27 was classified with the severe neonatal form because, at birth, she presented severe hypotonia and weakness, multiple bone fractures, scoliosis, arthrogryposis and ventilatory insufficiency. This patient needed NIV after birth and gastrostomy. At one year of age, she had complete head control and was able to sit with support and maintained the use of NIV. She started using pyridostigmine during the first year of life associated with multidisciplinary treatment.

Patient 30 also presented with the severe form and showed severe hypotonia and weakness in the face and limb muscles associated with multiple arthrogryposis since birth. She started using NIV at two years after multiple pneumonias. She was able to sit at two years of age. At the age of 5 years, she is still not able to stand up or walk independently, and she has scoliosis. In this patient, according to the laboratory report (Invitae), the mutations were also present in different alleles.

### 2.3. Muscle Biopsy

The slides from the muscle biopsy of each patient were reviewed, and the data are provided in Table 2.

In 28 patients, muscle biopsy was performed and showed rods inside the muscle fibers. Type I predominance was present in all patients, and in some, there was total predominance.

In two patients (P24 and P29), muscle biopsy was not performed; P24 had a sibling who had already had a muscle biopsy, and in P29, it was not necessary because the molecular diagnosis was conclusive.

### 2.4. Muscle Imaging

Muscle imaging of the lower limbs was performed in eight patients (P2, P3, P7, P14, P15, P16, P17, P21) and whole-body MRI in one patient (P6) (Table 2).

Among the patients with nebulin mutations, muscle imaging was performed in five cases (P14, P15, P16, P17, P21). In P16, muscle tomography was performed because she could not tolerate the supine position for a long time. This patient showed a diffuse and moderate involvement of thigh muscles and accentuated fat replacement in the legs (Figure 2).

In case 21 with a severe weakness, we observed moderate involvement of the thigh muscles (the vastus lateralis, recto femoris and hamstring muscles) and leg muscles (anterior tibial, posterior, soleus, gastrocnemius and fibular) (Table 2). In the three patients with the typical form (P14, P15, P17), the predominant involvement was in the anterior tibial and soleus muscles, while the thigh muscles were spared.

Two patients (P2 and P3) with *ACTA1* mutations were submitted for muscle MRI. P3, with a severe clinical form, presented accentuated and diffuse involvement in the thigh and the leg. In P2, with the typical form, we observed a mild involvement of the anterior tibialis muscle with no abnormalities in the thigh.

In P6 with the *TPM2* mutation, the whole-body MRI showed facial involvement, mainly temporal and lateral pterygoid muscles, and distal involvement in the lower legs, mainly soleus and flexor digitorum muscles (Figure 3).

For P7 with a *TPM3* mutation, the muscle MRI showed fat infiltration in the thigh muscles, especially the sartorius and adductor magnus, and a predominant involvement of the anterior compartment of the leg (Figure 3).

## 3. Discussion

Nemaline myopathies are considered among the most frequent congenital myopathies worldwide [2,44]. In Brazil, a large country with an estimated population of over 203.6 million, which is dispersed in a huge area, we still do not have an estimate of the congenital myopathies prevalence and of the different subgroups’ distribution.

Over the last 20 years, we have been evaluating Brazilian NM patients, characterizing their clinical phenotype and morphological and molecular aspects [45,46,47]. In the present study, we included data of NM patients from different states of Brazil, including Minas Gerais, São Paulo and Bahia. The molecular techniques changed over the years, varying from Sanger sequencing, next-generation sequencing through customized panels for neuromuscular disorders or whole-exome analysis associated with tools for identifying CNVs [27,28,29]. This has also significantly modified the capacity to identify mutations in so many genes, including a substantial one (*NEB*) involved in NM.

According to several studies still using the old clinical classification, the predominant NM forms are the typical, followed by the intermediate form, and the severe and childhood/juvenile forms [2,3,27]. In the present study, we used the new classification and found twenty-four patients with the typical form, three with the severe form and three with the mild form [5]. Additionally, concordant with the literature, we observed phenotype variability considering the same gene, especially with the nebulin and *ACTA1* genes [27,35,44,48].

During the follow-up of 20 NM patients, we observed the importance of monitoring the respiratory function, since all patients with severe forms, as well as a proportion of patients with the typical form, needed ventilatory support. Among the twenty patients with the typical form, three died because of respiratory complications, and six needed NIV despite preserving appendicular muscle strength and the ability to walk. This pattern of respiratory involvement out of proportion with the appendicular skeletal muscles is common in NM, which can mask the risk of respiratory complications in the follow-up of these patients [27,49,50]. This finding can be explained by histopathological studies, which showed the presence of nemaline bodies in the intercostal and diaphragmatic muscles, sometimes in higher quantity than in the appendicular skeletal muscles [51].

The histologic hallmark of NM is the presence of nemaline bodies or rods inside the muscle fiber [4]. The predominance of type 1 fibers, defined as more than 55% of type 1 fibers, is a common finding in NM, and in some patients, it can be seen as a complete type 1 uniformity [39,47]. In the present study, we observed the presence of rods and type 1 predominance in all patients who underwent muscle biopsy.

Genetically, NM is very heterogeneous, including more than 12 genes to date [5]. The most common causes of NM are recessive mutations in *NEB* and de novo dominant mutations in *ACTA1* [2,27,35]. Considering the families included in this study, we found sixteen families with nebulin mutations (64%), five with *ACTA1* mutations (20%), two families with *KLHL40* mutations (8%) and one family with mutations in *TPM2* (4%) and *TPM3* (4%). Therefore, we were able to genetically classify all 25 families, which is a great advance, achieved mainly because of the possibility to use more powerful molecular tools.

### 3.1. Nebulin Gene Mutations

Considering the clinical manifestations, our findings are in accordance with the literature, since the most common clinical form related to nebulin mutations in our patients was the typical form, observed in 17/20 of our patients, followed by the mild childhood form (3/20) [3,5,27].

The molecular screening of variants in the nebulin gene is very challenging, since this gene contains 183 exons, and in the middle of the gene, there is a triplicated region (TRI) consisting of a perfectly repetitive series of 8 exons (82–89, 90–97 and 98–105), and there are three regions with alternatively spliced exons (63–66, 143–144 and 167–177) [52,53]. NGS significantly improved the capacity to identify mutations in *NEB*, but in some cases, it can be insufficient, and it would be necessary to combine additional techniques, such as CGH microarray and RNA analysis [29]. In the present study, all patients with nebulin mutations showed biallelic mutations (P9–P28). Initially, in two patients (P21 and P22), we found only one mutation. However, we used the next-generation software for CNVs and/or the CGH array (Motor Chip v3) and could confirm the presence of two CNVs. The first CNV identified (P21) was a duplication of exons 82 to 105, which is located in the triplicated region of the *NEB* gene (TRI) and is frequently difficult to analyze in exome sequencing. In 2016, Kiiski et al. [28] used the NM-CGH microarray in 196 NM families and identified *NEB* TRI variation in 13% of the families and in 6% of the controls. They identified that the *NEB* gene tolerates deviation of one TRI copy, but two or more might be pathogenic. So, in our patient P21, we suggest that the CNV could be pathogenic. The exon 29 deletion (P22) was observed using the software XHNM but not when the Motor Chip v3 was used. This last mutation needs to be better studied.

The prevalence of different types of mutations in nebulin was estimated as follows: splice site mutations (34%), frameshift mutations (32%), small (<20 bp) deletions or insertions, nonsense mutations (23%) and missense mutations (7%). Large deletions and duplications (>1 Kb) were considered rare (4%) [27]. Similarly, in our cohort, we found that splice site and frameshift variants were the more frequent types of mutations. Importantly, in all *NEB* patients, at least one truncating mutation was found (frameshift, splice site or stop-gain).

Additionally, we identified eleven novel variants in Brazilian patients: eight were truncating, two CNVs and one non-frameshift duplication (VUS). In four patients, a novel truncating variant was associated with another previously described variant (P9, P10, P15, P26), and in two siblings, a novel truncating variant was found in homozygosity (P11/P12). So, in these five families, the genetic defect could probably lead to protein truncation or degradation. Among the identified CNVs, the duplication of exons 82–105 (P21) was associated with a previously described truncating mutation, and the segregation study showed this is a “de novo CNV”. The novel 29 exon deletion (P22) was also associated with a novel truncating mutation, and both were classified as pathogenic. In patient 13, the non-frameshift duplication, classified as VUS, was in association with a previously described truncating variant. However, in these last two patients, we could not perform a segregation study, and we need more studies to confirm the pathogenicity and consequence of mutation associations.

P20 was previously described in 2002, when we observed a lack of the C-terminal domain of the nebulin protein (antibody SH3) through Western Blot analysis in the muscle tissue of this patient [46]. In the present study, we could identify her molecular abnormality, which consisted of two novel truncating mutations in the C-terminal region of *NEB*: one frameshift mutation in exon 168 associated with a splice site mutation in exon 182, which could damage the SH3 domain. Similarly, a patient with a homozygous mutation in exon 185, which showed an absence of labeling with the SH3 antibody, was also described in 1999 [9]. On the other hand, in the *NEB*–SH3 mouse model, generated by knocking in a nonsense codon in the last exon, eliminating the C-terminal SH3 domain of nebulin, no nebulin expression abnormalities or altered localization of the interaction partners of the SH3 domain were observed [54]. The mouse presented a mild phenotype and was more susceptible to eccentric contraction-induced injury. Based on these data, it was suggested that the SH3 deletion in mouse could not be as detrimental as in human, similar to the milder phenotype observed in the mdx mouse model of Duchenne muscular dystrophy [54]. It is important to notice, however, that both our patient and the patient described by Pelin et al., 1999 [9], presented mutations that can impair the SH3 region but also the serine-rich domain, which can be responsible for the clinical manifestation of the disease.

The most frequent mutation described in different populations is the p.Arg2478_Asp2512del deletion encompassing exon 55 (del55) [55]. Among our patients, we did not find the del55 mutation in any patient. On the other hand, we identified the nebulin mutation c.246843G>C in three unrelated patients (P14, P16 and P17). In fact, this mutation has been observed in other studies, including in one French family [39] and in two Korean patients [56]. Our three patients are from a region with many consanguineous marriages, and a common origin cannot be ruled out for this mutation.

### 3.2. ACTA1 Gene Mutations

The *ACTA1* gene encodes the skeletal muscle alpha-actin protein, which is the predominant actin isoform in the sarcomeric thin filament. Mutations in *ACTA1* are the second most frequent cause of nemaline myopathy. More than 177 mutations were described in the *ACTA1* gene, including missense, nonsense, frameshift and splice site mutations. Most mutations are dominant and de novo; only 10% of the pathogenic variations are recessive. In our cohort, we identified five families with *ACTA1* heterozygous mutations in sporadic cases. Four patients were clinically classified with the typical form and one with the severe form. Considering the type of mutations, four of them were missense mutations previously described in the literature [34,35].

One novel intronic splice site variant (intron 2) was found in P2 and was classified as a VUS. A variant in the same nucleotide (c.130-5T>C) was classified as benign due to the high frequency in two databases (GNOMAD, LOVD). However, the variant identified in our patient (c.130-5T>A) was analyzed using in silico prediction software, such as Human Splicing finder and MaxEntScan, and was also absent in the parents. This might indicate that this variant has a pathogenic effect but requires additional studies to fully prove its pathogenicity.

A severe clinical phenotype with the presence of intranuclear rods has been associated with variants in *ACTA1* located in a cluster between amino acid residues 139 and 165 [35]. P3 in this study showed a severe form of NM, needing ventilatory support and gastrostomy in the neonatal period associated with a heterozygous mutation in exon 4, which is located inside the cluster between amino acid residues 139 and 165. Although ultrastructural analysis was not available in this case, to look for intranuclear rods, a phenotype–genotype correlation would be compatible with the literature in this case.

Our P3 presented an asymmetric weakness distribution, with a more intense weakness on the left side of the body and on the right side of the face. The cause of this asymmetry is not clear. Interestingly, two patients with asymmetric weakness due to a mosaic pattern of mutations in the *ACTA1* gene have been described by Lonarge et al., 2020 [57]. However, our P3 showed no molecular evidence of a mosaic pattern of the mutation, corroborated by his very severe phenotype.

### 3.3. Mutations in Other Genes Related to NM

Other genes have been related to NM in a small percentage of patients, including *TPM3*, *TPM2*, *TNNT1*, *KBTBD13*, *CFL2 (COFILIN2)*, *KLHL40*, *KLHL41*, *LMOD3*, *MYO18B*, *MYPN*, *RYR3*, *TTN3*, *ADSSL1*, Filamin C and *MYH2* [2,5,14,39]. They are associated with autosomal dominant, recessive or sporadic cases. Our NM family with mutations in the *TPM3* gene showed an autosomal dominant inheritance, with a father and a son presenting a proximal lower limb weakness. The son presented a severe progressive scoliosis requiring spine surgery. Our *TPM2*-related patient is a sporadic case, carrying the recurrent mutation p. K7del, presenting the typical NM form with limitation of the opening of the mouth [36].

Two of our patients with the severe NM form were compound heterozygous for the same two missense mutations (c.1405G>A; p.Gly469Ser and c.1498C>T; p.Arg500Cys) in *KLHL40*. This autosomal recessive nemaline myopathy (NEM8) was first described in 2013 in 20% of the patients with the very severe neonatal form, including fetal akinesia or hypokinesia, contractures, fractures and respiratory and swallowing difficulties apparent at birth. The average age at death was 5 months [14]. On the other hand, the first patient with a milder phenotype was described in 2016 [40], related to the mutation c.1498C>T; p.Arg500Cys in a homozygous state. Our patient P29 showed a phenotype compatible with the severe NEM8 form in the neonatal period. However, subsequently after birth, she was under the administration of pyridostigmine daily and with respiratory support (NIV), and her motor development improved, since she was able to sit with support at the age of 1,6 years. The other patient, P30, showed severe weakness and contractures at birth, also with motor improvement during the first three years of life. She was stable, receiving only supportive treatment. At five years of age, she is able to sit without support and using nocturnal NIV. The c.1405G>A; p.Gly469Ser mutation was previously associated with the severe phenotype, but the fact that mutation c.1498C>T; p.Arg500Cys was found in a milder patient suggests the possibility that the combination of these two mutations could explain their clinical improvement during these first years of life and longer survival.

Additionally, an interesting point is finding the same genotype in *KLHL40* in two unrelated families, since it is a very rare condition. The families are from the same state in Brazil (Sao Paulo), which is very large and has an ethnically heterogeneous population. However, we could not exclude a possible founder effect of these mutations or unknown relatedness between these two families.

### 3.4. Muscle Imaging in NM

Magnetic resonance imaging has been considered a powerful method for the study of muscle diseases. In nemaline myopathy, some patterns of muscle involvement are related to the mutated gene; however, a complete correlation between the gene and muscle commitment is not recognized yet [30].

In the present study, among the patients with nebulin mutations, we observed a pattern of selective muscle involvement related to the clinical severity of the disease, independently of age. In patients with the typical form (mild phenotype), the thigh muscles were spared, and the anterior tibial and soleus muscles were more affected in the leg. In patients with a more severe weakness, the thigh muscles were also affected, including the vastus lateralis, rectus femoris, hamstring muscles, and in the leg, mainly the anterior tibial and soleus muscles were involved, similar to previously described findings [30]. In one patient (P16) with a typical form but progressive distal weakness, we observed a diffuse involvement in the thigh and leg muscles but more intense in the legs, which was compatible with her distal muscle involvement.

In the literature, muscle MRI in patients with *ACTA1* mutations and the mild phenotype showed diffuse involvement in the thigh, mainly in the adductor magnus and vastus muscles, and in the legs, the tibial anterior and the soleus were more affected. We could perform muscle MRI in two patients with mutations in *ACTA1*, one with a typical form and mild weakness and the other with a severe form and accentuated weakness. In both patients, we observed that the pattern of fat infiltration in muscle MRI was related to the severity of the clinical phenotype.

Considering the patient with a *TPM2* mutation, using whole-body MRI, we could confirm the classical pattern of muscle involvement, with the predominant alterations in masticatory and distal leg muscles [31].

Data on muscle imaging in *TPM3*-related myopathy are scarce. In 2014, Schreckenbach et al. [58] published data from a German family with cap myopathy and mutations in *TPM3*. The muscle MRI showed a more intense involvement of the posterior compartment in the thigh and in the legs; the more affected muscles were the tibial anterior and soleus. Differently, in our patient, the thigh muscles were mildly affected, more intensely in the sartorius and adductor magnus muscles, while in the legs, the anterior compartment was more involved.

## 4. Materials and Methods

### 4.1. Patients

Patients with NM were evaluated over the last 20 years in two reference centers for NMD in Sao Paulo and Minas Gerais. Additional patients were referred from several other states of the country.

The study included a total of 30 patients from 25 apparently unrelated Brazilian families. Patients were clinically evaluated, and some of them were followed for months to years by the same pediatric neurologist and multidisciplinary team. Based on clinical disease severity, the patients were classified into six groups: severe, intermediate and typical congenital forms, a mild childhood/juvenile-onset form, adult-onset NM and atypical forms.

### 4.2. Muscle Biopsy

Muscle biopsy was performed for diagnostic purposes. Cryostat sections of 8 μm were submitted to conventional histological and histochemical techniques: hematoxylin and eosin (H&E), modified Gomori trichrome (GO), periodic acid Schiff technique (PAS), Oil Red O (ORO), reduced nicotinamide adenine dinucleotide dehydrogenase-tetrazolium reductase (NADH-TR), succinic dehydrogenase (SDH), cytochrome c oxidase (COX) and adenosine triphosphatase (ATPase) pre-incubated at pH 9.4, 4.63 and 4.35.

The presence of rods was analyzed in the GO stain and multiminicores or cores on NADH reaction. Type I predominance was considered when the proportion of type I fibers was more than 55% of the total fibers in quadriceps and more than 60% in deltoid muscles [40,48]. There was no sample available for electron microscopy.

### 4.3. Mutation Analysis

Genomic DNA was extracted from blood using standard protocols.

Next-generation sequencing was performed using a customized NGS panel with 95 genes involved in neuromuscular disorders (NMD) or whole-exome sequencing using Illumina’s Nextera kits for library preparation and custom capture of these genes or Agilent 44 Mv2 SureSelect Exon enrichment kit. Sequencing was performed on an Illumina MiSeq sequencer or Hiseq 2000. After filtering, the variants were compared to the control populations of the 1000-genome (NIH) and 6500-exome sequencing project from the Washington University and the recently created Online Archive of Brazilian Mutations—AbraOM [59] (http://www.abraom.ib.usp.br/, accessed on 22 August 2022). Rare variants, mainly in the *ACTA1*, *TPM2* and *TPM3* genes, were selected and analyzed using bioinformatics tools. Gene mutation databases HGMD, LOVD and ClinVar were consulted in search of previously described pathogenic variants. The American College of Medical Genetics and Genomics (ACMG) pathogenicity classification guidelines [60] were used for variant classifications.

The pathogenicity of de novo variants was analyzed on prediction sites, including: MutationTaster, Predict SNP1, CADD, DANN, FATHMM, FunSeq2, GWAVA, VEP, SIFT, Polyphen1, Polyphen2 and Human splicing finder3.0. Some of the patients were previously studied through Sanger sequencing.

In patients in whom we found only one mutation, we looked for deletions or duplications using the Next Gene Software/XHMM Software (XHMM (eXome-Hidden Markov Model, 2019) and through Motor Chip v3, a genomic hybridization microarray for copy number mutations in 245 genes related to neuromuscular disorders developed by Vincenzo Nigro’s Lab [41].

Mutations are reported according to the cDNA reference sequences of nebulin (NM_001164507.1), *ACTA1* (NM_001100), *TPM2* (NM_003289.4), *TPM3* (NM_152263.4), *KLHL40* (NM 152393.4).

We confirmed the inheritance in four families by Sanger sequencing samples from the parents.

### 4.4. Muscle Imaging

Muscle imaging was performed in patients able to collaborate without sedation. The imaging protocol was performed following the standard sequences: Axial T1-weighted (turbo/fast) spin echo, Axial T2-weighted (turbo/fast) spin echo with fat suppression. The slice thickness was 5–8 mm with interslice gap of 1–2 mm and in-plane resolution of 1–2 mm. We studied the lower extremities or the whole body.

Muscle tomography of the lower limb was performed in patients who could not tolerate the recumbent position.

For the image analysis, we used the semi-quantitative visual rating scale described by Mercuri et al. in 2002 and Wattjes et al. in 2010 [42,43].

## 5. Conclusions

The molecular analysis in the present study showed that mutations in the *NEB* gene were the most common cause of NM, followed by mutations in the *ACTA1* gene, in this cohort of Brazilian patients. In total, 34 different variants were identified in this cohort, 12 of them being newly described mutations. The *NEB* mutation c.24579 G>C was recurrent in three unrelated patients, but from the same state of Brazil, with a high frequency of consanguinity, suggesting a common ancestor. Two patients with severe forms of the disease presented the same *KLHL40* mutations in compound heterozygosity, although their families were not related.

The most frequent form of NM was the “typical form”, which can be caused by mutations in different genes, and phenotypic variability can be observed among patients with mutations in the same gene. Respiratory involvement was very common in NM patients and could be out of proportion with limb weakness. This aspect is very important and needs special attention in the follow-up of these patients.

Muscle MRI could show different patterns of muscle involvement associated with the affected gene, which can be useful in highlighting this gene for a more detailed analysis during the NGS screening for mutations. On the other hand, considering the same gene, this pattern can be variable with the severity of clinical presentation.

Considering the high frequency of *NEB* mutations and the complexity of this gene, NGS tools should be combined with techniques designed to detect CNVs, especially in patients with only one identified mutation.

## Figures and Tables

**Figure 1 ijms-23-11995-f001:**
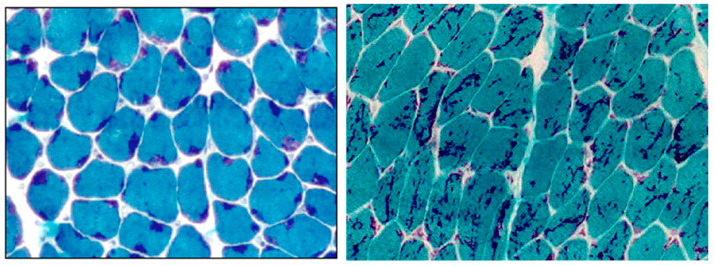
Muscle biopsy showing the presence and distribution of nemaline bodies inside myofibers: on the left picture, from P1, there are predominantly large rods in subsarcolemmal region; on the right picture, from P20, there are predominantly small intermyofibrillar rods (magnification ×20).

**Figure 2 ijms-23-11995-f002:**
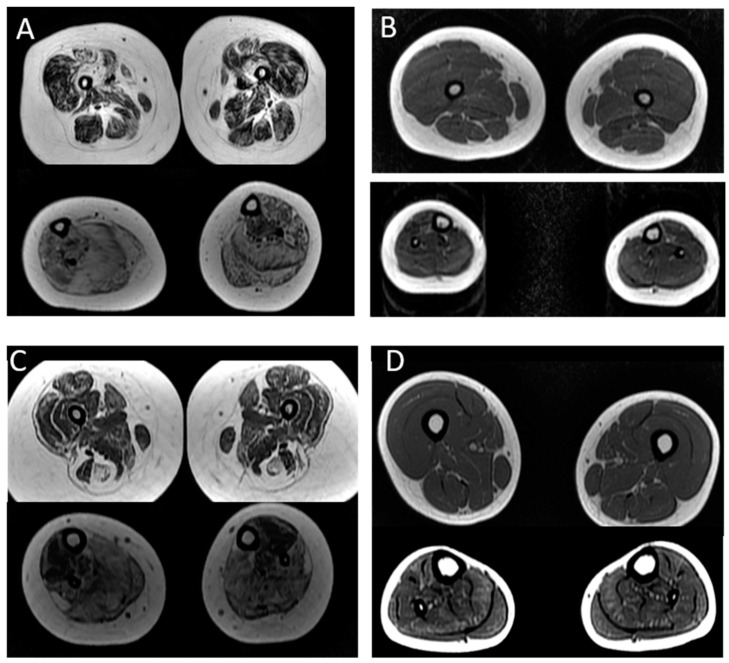
MRI pattern in patients with mutations in the *ACTA1* and *NEB* genes: (**A**,**B**) patients with mutations in the *ACTA1* gene. In (**A**), there is a diffuse infiltration in the thigh and legs in P3 (severe form); in (**B**), there is isolate involvement of tibial anterior in the legs in P2 (typical form). (**C**,**D**) patients with nebulin mutations; in (**C**), we observe a diffuse infiltration in the thigh and legs in a patient with severe weakness who never walked (P21); In (**D**), we observe that the predominant involvement was in the anterior tibial and soleus muscles, while the thigh muscles were spared (P15).

**Figure 3 ijms-23-11995-f003:**
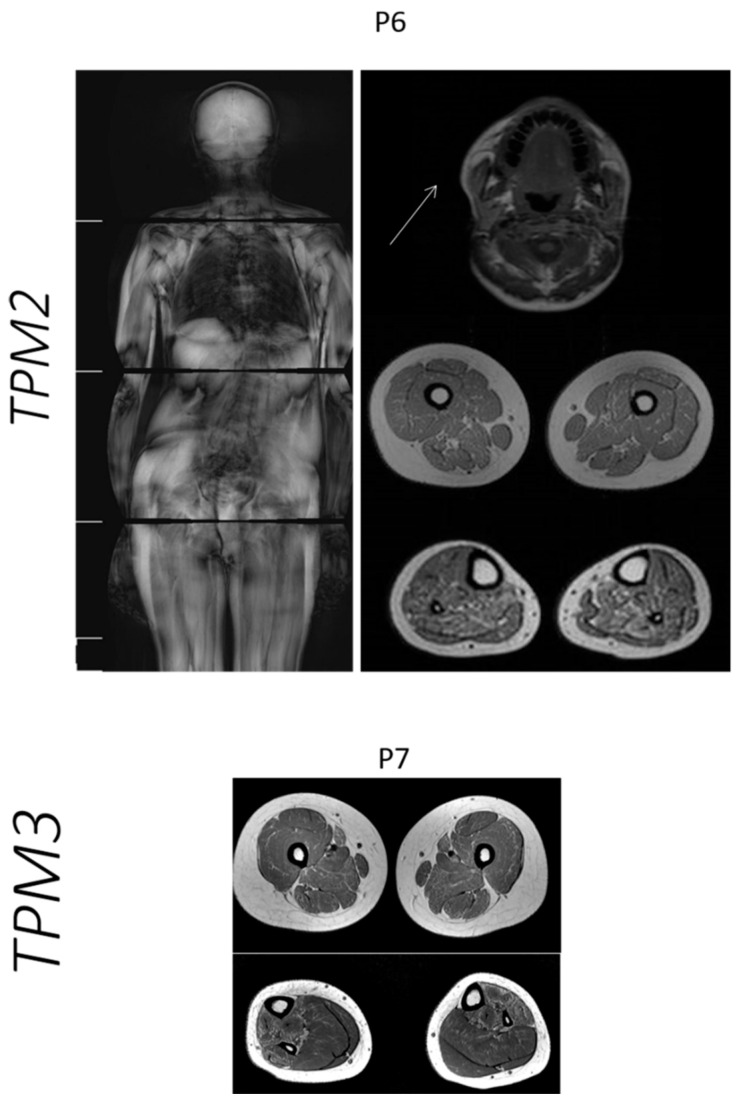
MRI pattern in patients with mutations in the *TPM2* and *TPM3* genes. In P6 with *TPM2* mutation, the whole-body MRI showed facial involvement, mainly temporal and lateral pterygoid muscles, and distal involvement in lower legs. In P7 with *TPM3*, the muscle MRI showed fat infiltration in thigh muscles, especially sartorius and adductor magnus, and predominant involvement of the anterior compartment of the leg.

**Table 1 ijms-23-11995-t001:** Clinical and molecular data of the patients. Abbreviations: ACMG: American College of Medical Genetics and Genomics; NIV: Non-invasive ventilation; VUS: Variant of uncertain significance.

Case, Sex	Family	Clinical Phenotype	Molecular Analysis			
		Age at Last Evaluation and Clinical Findings	Gene	Mutations	Type/Segregation Parental	ACMG 6/2022 Classification
1 Male	Family 1	Typical 20 years: Not Ambulant since 11 years of age NIV at night, Scoliosis	*ACTA1*	NM_001100.4:exon4:c.541G>C: p.(Asp181His)	Missense heterozygous Previously described [34]	Likely Pathogenic PM2 PP2 PP3 PS4_Supporting PP4
2 Male	Family 2	Typical 13 years: Ambulant No VNI, No Scoliosis	*ACTA1*	NM_001100.4:c.130-5T>A	Splice site Heterozygous Novel Not present in the parents	VUS PM2 PP2 PP3 PP4
3 Male	Family 3	Severe 15 years: Not Ambulant (never) Permanent Invasive ventilation (tracheostomy) Gastrostomy, Scoliosis	*ACTA1*	NM_001100.4:exon4:c.478G>A: p.(Gly160Ser)	Missense heterozygous Previously described [35]	Likely Pathogenic PM2 PP2 PP3 PS4_Supporting PP4
4 Male	Family 4	Typical 3 years: Ambulant No VNI/ No scoliosis	*ACTA1*	NM_001100.4:c.478G>A: p.(Arg198His)	Missense heterozygous Previously described [35]	Likely Pathogenic PM2 PP2 PP3 PS4_Supporting PP4
5 Female	Family 5	Typical 14 years: Ambulant No VNI/ No scoliosis	*ACTA1*	NM_001100.4:c.854T>G: p.(Met283Arg)	Missense heterozygous Previously described [35]	Likely Pathogenic PM2 PP2 PP3 PS4_Supporting PP4
6 Male	Family 6	Typical 13 years: Ambulant No VNI/ Mild scoliosis	*TPM2*	NM_003289.4:c.20_22 del: p.(Lys7del)	Frameshift heterozygous Previously described [36]	Pathogenic PM2 PS4 PP3 PM6
7 Male	Family 7 AD	Typical 19 years: Ambulant/Nocturnal VNI/Severe scoliosis	*TPM3*	NM_152263.4:c.502C>T: p.(Arg168Cys)	Missense heterozygous Previously described [37]	Pathogenic PM2 PS4 PM5 PM6
8 Male (father)		Typical 62 years: Ambulant with support Permanent VNI (>16 h/day), Scoliosis		NM_152263.4:c.502C>T: p.(Arg168Cys)		
9 Female	Family 8	Mild 27 years: Ambulant No VNI	*NEB*	NM_001164507.1:c.8501delA: p.(Lys2834ArgfsTer28)	Frameshift heterozygous Novel	Pathogenic PVS1 PM2 PP4
				NM_001164507.1:c.1674+2T>C	Splice site heterozygous Previously described [27]	Pathogenic PVS1 PM2 PS4_Supporting PP4
10 Male	Family 9	Typical Dead at 25 years old: Not ambulant Permanent invasive ventilation (tracheostomy) Severe scoliosis	*NEB*	NM_001164507.1:exon170:c.24189_24192dup p.(Glu8065SerfsTer5)	Frameshift heterozygous Previously described [38]	Pathogenic PVS1 PM2 PS4_Supporting PP4 PP4
				NM_001164507.1:exon134:c.20466+2T>A	Splice site heterozygous Novel	Pathogenic PVS1 PM2 PM3_Supporting PP4
11 Female	Family 10 consanguineous	Typical 4 years: Ambulant, No VNI	*NEB*	NM_001164507.1:exon172:c.24304_24305dup: p.(Leu8102PhefsTer44)	Frameshift homozygous Novel	Pathogenic PVS1 PM2 PP4
12 Male (brother)		Typical 6 years: Ambulant, No VNI		NM_001164507.1:exon172:c.24304_24305dup: p.(Leu8102PhefsTer44)	Frameshift homozygous Novel	Pathogenic PVS1 PM2 PP4
13 Male	Family 11	Typical 14 years: Ambulant/ No VNI	*NEB*	NM_001164507.1:exon 63 c.8889+1G>A p.(?)	Splice site heterozygous Previously described [27]	Pathogenic PVS1 PM2 PS4_Moderate PP4
				NM_001164507.1:c.6869_6870insTGC: p.(Ala2290dup)	Non-frameshift duplication heterozygous Novel	VUS PM2 PM3_Supporting PP4
14 Female	Family 12	Typical 19 years: Ambulant Nocturnal NIV Scoliosis	*NEB*	NM_001164507.1:exon151:c.22170G>A: (p.Tyr7390Ter	Stop-gain Heterozygous Previously described [38]/maternal	Pathogenic PVS1 PM2 PS4_Supporting PM3_Supporting PP4
				NM_001164507.1:exon174: c.24579G>C; p.(Ser8193Ser)	Splicing heterozygous Previously described [39]/paternal	Likely Pathogenic PM2 PS4_Supporting PP3 PM3_Supporting PP4
15 Female	Family 13	Typical 23 years: Ambulant Permanent NIV (> 16 h a day) Scoliosis	*NEB*	NM_001164507.1:exon176:c.24735_24736del: p.(Arg8245fsTer2)	Frameshift deletion heterozygous Previously described [27,39]	Pathogenic PVS1 PM2 PS4_Supporting PP4
				NEB_NM_001164507.1:IVS122:c.19102-8_19102-4del	Splice site heterozygous Novel	VUS PM2 PP4
16 Female	Family 14	Typical 25 years: Lost the ambulation at 17 years of age Permanent NIV (>16 h a day) Severe Scoliosis	*NEB*	NM_001164507.1:exon174: c.24579G>C; p.(Ser8193Ser)	Splice site heterozygous Previously described [39]/paternal	Likely Pathogenic PM2 PS4_Supporting PP3 PM3_Supporting PP4
				NM_001164507.1:exon48:c.6078delA p.(Lys2026fs)	Frameshift heterozygous Previously described [26]/Maternal	Pathogenic PVS1, PM2 PS4_Supporting PP4 PM3_Supporting
17 Female	Family 15	Typical 20 years: Ambulant Nocturnal NIV Scoliosis	*NEB*	NM_001164507.1:exon151:c.22170G>A: p.(Tyr7390Ter)	Stop-gain heterozygous Previously described [38]—nonsense mutation/paternal	Pathogenic PVS1 PM2 PS4_Supporting PP4
				NM_001164507.1:exon174: c.24579G>C; p.(Ser8193Ser)	Splice site heterozygous Previously described [39]/maternal	Likely Pathogenic PM2 PS4_Supporting PP3 PM3_Supporting PP4
18 Female	Family 16	Typical 4 years: Ambulant No NIV	*NEB*	NM_001164507.1: exon173: c.23601_23602del: p.(Lys7867AsnfsTer3)	Frameshift heterozygous NM consortium Previously described [38]	Pathogenic PVS1 PM2 PP4
19 Female (sister)		Typical 2 years: Ambulant No NIV		NM_001164507.1Int28:c.2835+5G>C	Splice site heterozygous NM consortium	VUS PM2 PP4
20 Female	Family 17	Typical Dead at 15 years: Ambulant No NIV Acute respiratory failure	*NEB*	NM_001164507.1:c.23878_23881dup p.(Thr7961AsnfsTer16)	Frameshift heterozygous Novel	Pathogenic PVS1 PM2 PP4
				NM_001164507.1:c.25405-1G>C	Splice site heterozygous Novel	Pathogenic PVS1 P M2 PP4
21 Female	Family 18	Typical 12 years: Not ambulant Permanent NIV (had tracheostomy) Gastrostomy Scoliosis	*NEB*	NM_001164507.1:c.1623delT: p.(Asp541IlefsTer15)	Frameshift heterozygous Previously described [26]/maternal	Pathogenic PVS1 PM2 PS4_Supporting PP4
				Motor Chip (V. Nigro): duplicação 82 a 105	CNV heterozygous Novel	
22 Male	Family 19	Typical 13 years: Ambulant No NIV	*NEB*	NM_001164507.1:exon34:c.3648del p.(Lys1218ArgfsTer6)	Frameshift heterozygous Novel	Pathogenic PVS1 PM2 PP4
				del exon 29	CNV Novel Not confirmed by Motor Chip	Pathogenic PVS1 PM2 PP4
23 Male	Family 20	Mild 27 years: Ambulant No VNI	*NEB*	NM_001164507:exon43:c.5343+5G>A	Splicing homozygous Previously described [26]	Likely pathogenic PM2 (rara) PS4_sup PM4, PM3_sup, PP4
24 Female (sister)		Mild 25 years: Ambulant No VNI		NM_001164507:exon43:c.5343+5G>A	Splicing homozygous Previously described [26]	Likely pathogenic PM2 (rara) PS4_sup PM4, PM3_sup, PP4
25 Male	Family 21	Typical 8 years: Not ambulant (never walked) Progressive weakness (loss of head control and the capacity to sit) Permanent NIV (>16 h a day) Gastrostomy Severe scoliosis	*NEB*	NM_001164507.1:c.21076C>T p.Arg7026Ter	Stop-gain heterozygous Previously reported [27]	Pathogenic PVS1 PM2 PS4_Moderate PP4
				NM_001164507.1:c.24192_24193insTCAA p.Glu8065Serfs5	Frameshift heterozygous Previously reported [27]	Pathogenic PVS1 PM2 PS4 PM3 PP4
26 Male	Family 22	Typical Dead at 5 years old: Ambulant No NIV (Acute respiratory failure)	*NEB*	NM_001164507.1:exon129:c.19944G>A: p.(Ser6648Ser)	Splice site heterozygous Previously reported [27]	Likely Pathogenic PM2 PS4 PP3 PP4
				NM_001164507.1:exon105:c.16423A>T: p.(Lys5475Ter)	Stop-gain heterozygous Novel	Pathogenic PVS1 PM2 PP4
27 Female	Family 23	Typical 18 years: Ambulant No NIV No scoliosis	*NEB*	NM_001164507:exon43:c.5343+5G>A	Splice site heterozygous Previously reported [26]	Likely Pathogenic PM2 PS4_sup PM4 PM3_sup PP4
28 Male (brother)		Typical 13 years: Ambulant No NIV No scoliosis		NEB:NM_001164507:exon29:c.2943G>A p.Glu981Glu	Splice site heterozygous Previously reported by Invitae on Clinvar rs398124170	Likely pathogenic PM2 PS4_sup PP3 PM3 PP4
29 Female	Family 24	Severe 1,5 years: sits with support Nocturnal VNI Scoliosis	*KLHL40*	NM_152393.4:c.1405G>A: p.(Gly469Ser)	Missense heterozygous Previously reported [14]/paternal	VUS PM2 PP3 PP4
				NM_152393.4:c.1498C>T: p.(Arg500Cys)	Missense heterozygous Previously reported [40]/maternal	VUS PM2 PS4_Supporting PP3 PP4
30 Female	Family 25	Severe 5 years: Not ambulant Distal Arthrogryposis Nocturnal VNI Scoliosis	KLHL40	NM_152393.4:c.1405G>A: p.(Gly469Ser)	Missense heterozygous Previously reported [14]	VUS PM2 PP3 PP4
				NM_152393.4:c.1498C>T: p.(Arg500Cys)	Missense heterozygous Previously reported [40]	VUS PM2 PS4_Supporting PP3 PP4

**Table 2 ijms-23-11995-t002:** Muscle MRI: Pattern of muscle involvement in nemaline patients.

Case	Age at Muscle MRI	Gene	Muscle MRI or CT (Mercuri Grade *)	
			Thigh:	Legs:
P2	4 years	*ACTA1*	No involvement	Tibialis anterior muscle: grade IIa Soleus muscle: no involvement
P3	8 years	*ACTA1*	Diffuse fat infiltration of thigh muscles: grade III Biceps femoris and vastus lateralis muscles: grade IIb	Diffuse fat infiltration: grade III Tibialis posterior muscle: grade IIb
P6	12 years	*TPM2*	Vastus lateralis and rectus femoris muscles: grade I OBS: Axial T1 image of the head: Involvement of temporal, masseter and pterygoid lateral muscles: grade II	Soleus muscle: grade IIb Gastrocnemius muscle: grade IIa
P7	13 years	*TPM3*	Sartorius muscle: grade IIb Adductor magnus muscle: grade IIa Hamstring muscles: grade I	Soleus muscles: grade I Gastrocnemius muscle: grade I Tibialis anterior muscle: grade IIb Tibialis posterior muscle: grade IIb
P14	17 years	*NEB*	Diffuse fat infiltration of thigh muscles: grade I	Tibialis anterior muscle: grade III Soleus muscle: grade IIa
P15	15 years	*NEB*	Diffuse fat infiltration of thigh muscles: grade I	Tibialis anterior muscle: grade IIa Soleus muscle: grade IIb Fibular muscle: grade IIa
P16	17 years	*NEB*	Adductor longus muscle: grade III Adductor magnus muscle: grade III Rectus femoris muscle: grade IIa Vastus lateralis muscle: grade IIa	Tibialis anterior muscle: grade IV Soleus muscle: grade IV Fibular muscle: grade IV
P17	14 years	*NEB*	Diffuse fat infiltration of thigh muscles: grade I	Tibialis anterior muscle: grade II Soleus muscle: grade II Fibular. muscle: grade II
P21	7 years	*NEB*	Adductor longus muscle: grade I Adductor magnus muscle: grade IIb Rectus femoris muscle: grade IIb Vastus lateralis muscle: grade IIb	Tibialis anterior muscle: grade IIa Tibialis posterior muscle: grade IIa Soleus muscle: grade III Gastrocnemius muscle: grade III Fibular muscle: grade III

* Semi-quantitative visual rating scale described by Mercuri et al., 2002 and Watjjes et al., 2010 [42,43].

## Data Availability

All data generated or analyzed during this study are included in this published article.

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
