# Peer review of "Nemaline Myopathy in Brazilian Patients: Molecular and Clinical Characterization"

_ijms, 2022, doi:10.3390/ijms231911995_

Round 1
Reviewer 1 Report
The authors present detailed molecular and clinical data for 30 patients from 25 unrelated Brazilian families, all of whom have nemaline myopathy. As such this detail provides a valuable contribution to the literature.
Unfortunately there are a number of typographical errors. Many errors exist in spacing eg: 1. Introduction is on p1 with all content on p2; trivial spacing eg: families: 10,16,20, 23 etc
Table 1: consider layout of the content of cells especially column 3 Clinical phenotype and column 6 Type
eg Not Ambulant since 11 years of age NIV at night Scoliosis
the abbreviation VNI is not described anywhere in the article
P14 check brackets in column Type/parental segregation
P17 is listed as 20 anos
Table 2:
Inconsistency in the naming of muscles and spelling
Alignment of text in some cells requires correction
English
Whilst in general the English is good, there are a number of occasions where it is not quite correct eg They did not presented scoliosis
Abbreviations are used without explanation eg CM used in the discussion
Author Response
Reviwer 1
The authors present detailed molecular and clinical data for 30 patients from 25 unrelated Brazilian families, all of whom have nemaline myopathy. As such this detail provides a valuable contribution to the literature.
Unfortunately there are a number of typographical errors. Many errors exist in spacing eg: 1. Introduction is on p1 with all content on p2; trivial spacing eg: families: 10,16,20, 23 etc
Corrections were made.
Table 1: consider layout of the content of cells especially column 3 Clinical phenotype and column 6 Type
eg Not Ambulant since 11 years of age NIV at night Scoliosis
the abbreviation VNI is not described anywhere in the article
P14 check brackets in column Type/parental segregation
P17 is listed as 20 anos
All corrected
Table 2:
Inconsistency in the naming of muscles and spelling
Tibialis anterior muscle ; soleus muscle; tibialis posterior muscle, Vastus lateralis muscle, rectus femoris muscle, gastrocnemius muscle, sartorius muscle, Adductor magnus muscle, Hamstrings muscles, fibular muscle, adductor longus muscle,
Alignment of text in some cells requires correction
corrected
English
Whilst in general the English is good, there are a number of occasions where it is not quite correct eg They did not presented scoliosis
corrected
Abbreviations are used without explanation eg CM used in the discussion
corrected
Reviewer 2 Report
The aim of this work was to screening mutations in Brazilian patients with nemaline myopathy. The effort to compile the molecular and clinical data of cohort of 30 Brazilian patients together should be commended. The manuscript should be of interest to the readership of the International Journal of Molecular Sciences. The logic of the article is clear however some parts of manuscript require carefully rearrangement.
I have the following comments:
1) Section 1: Introduction
In the third paragraph instead of: the genes encoding α-tropomyosin (TMP3) should be the genes encoding g-tropomyosin (slow α-tropomyosin ) (TPM3)
In the fourth and fifth paragraphs citation is wrong presented
e.g. to around 50% of the cases. [3]
including nemaline myopathy. [31–34].
2) Section 2: Results
Figure 1 has not include in the description the source of pictures.
Table 2 needs revision e.g.
- gene names should be written italic,
- ACMG 6/2022, PP2 PP3 PS4 abbreviations should be described,
- gender is written both in uppercase and lowercase
- Table 2 is filled with colors, unnecessarily
2.2.1-) Delete the additional signs. Should be 2.2.1 only.
In 2.2.1 section the beginning of the fourth and fifth paragraphs should be properly formatted.
Cut-off line is unnecessary.
2.2.2.-) Patients with mutations in the ACTA1 gene. Delete the additional signs. Should be 2.2.2 only.
In the first paragraph of section 2.4. the reference to table 2 should be change to Table 2.
Table 1 needs revison e.g.
- incorrectly writen gene names,
- grade written with uppercase or lowercase, y or Y too
Figure 2 description. Presenting full names of patients is inconsistent with the rest of text. Use P5 or P2 rather.
Figure 3 description needs rewriten gene names for italic format.
Section 3.1. Fourth paragraph is wrong formated.
Section Muscle imaging in NM should be numbered 3.4. Last acapits in this section are wrong formated.
Section 4: Materials and methods should be numbered as a last one.
Section 5: Conclusion should be presented before section Materials and methods.
Author Response
Reviewer 2
The aim of this work was to screening mutations in Brazilian patients with nemaline myopathy. The effort to compile the molecular and clinical data of cohort of 30 Brazilian patients together should be commended. The manuscript should be of interest to the readership of the International Journal of Molecular Sciences. The logic of the article is clear however some parts of manuscript require carefully rearrangement.
I have the following comments:
1) Section 1: Introduction
In the third paragraph instead of: the genes encoding α-tropomyosin (TMP3) should be the genes encoding g-tropomyosin (slow α-tropomyosin ) (TPM3)
corrected
In the fourth and fifth paragraphs citation is wrong presented
e.g. to around 50% of the cases. [3]
including nemaline myopathy. [31–34].
corrected
2) Section 2: Results
Figure 1 has not include in the description the source of pictures.
The figures corresponds to patients P1 and P20, and this information was included in the legend of the figure.
Table 2 needs revision e.g.
- gene names should be written italic,
- ACMG 6/2022, PP2 PP3 PS4 abbreviations should be described,
- gender is written both in uppercase and lowercase
- Table 2 is filled with colors, unnecessarily
Corrected. Table in color is to help to identify the involved gene mutation.
2.2.1-) Delete the additional signs. Should be 2.2.1 only.
In 2.2.1 section the beginning of the fourth and fifth paragraphs should be properly formatted.
ok
Cut-off line is unnecessary. - ok
2.2.2.-) Patients with mutations in the ACTA1 gene. Delete the additional signs. Should be 2.2.2 only.
All formation was revised
In the first paragraph of section 2.4. the reference to table 2 should be change to Table 2.
ok
Table 1 needs revison e.g.
- incorrectly writen gene names, OK
- grade written with uppercase or lowercase, y or Y too - OK
Figure 2 description. Presenting full names of patients is inconsistent with the rest of text. Use P5 or P2 rather.
Corrected
Figure 3 description needs rewriten gene names for italic format.
Corrected
Section 3.1. Fourth paragraph is wrong formated.
Corrected
Section Muscle imaging in NM should be numbered 3.4. Last acapits in this section are wrong formated.
Corrected
Section 4: Materials and methods should be numbered as a last one.
Section 5: Conclusion should be presented before section Materials and methods.
Corrected
Round 2
Reviewer 1 Report
The authors present additional information for the literature on the clinical presentation and genotype for a group of 30 patients with NM from 25 families
Please check for minor spelling errors once all corrections are accepted eg Table 2: P3 fat infiltration of fthigh muscles
Reviewer 2 Report
Revised manuscript is accepted in present form.